# ε-Polylysine Inhibits *Shewanella putrefaciens* with Membrane Disruption and Cell Damage

**DOI:** 10.3390/molecules24203727

**Published:** 2019-10-16

**Authors:** Weiqing LAN, Nannan ZHANG, Shucheng LIU, Mengling CHEN, Jing XIE

**Affiliations:** 1College of Food Science and Technology, Shanghai Ocean University, Shanghai 201306, China; wqlan@shou.edu.cn (W.L.); 15836066261@163.com (N.Z.); chenml5211314@126.com (M.C.); 2College of Food Science & Technology, Guangdong Provincial Key Laboratory of Aquatic Product Processing and Safety, Guangdong Ocean University, Zhanjiang 524088, China; Lsc771017@163.com; 3Shanghai Aquatic Products Processing and Storage Engineering Technology Research Center, Shanghai 201306, China; 4National Experimental Teaching Demonstration Center for Food Science and Engineering, Shanghai Ocean University, Shanghai 201306, China

**Keywords:** ε-polylysine (ε-PL), *S. putrefaciens*, membrane disruption, cell damage

## Abstract

ε-Polylysine (ε-PL) was studied for the growth inhibition of *Shewanella putrefaciens* (*S. putrefaciens*). The minimal inhibitory concentration (MIC) of ε-PL against *S. putrefaciens* was measured by the broth dilution method, while the membrane permeability and metabolism of *S. putrefaciens* were assessed after ε-PL treatment. Additionally, growth curves, the content of alkaline phosphatase (AKP), the electrical conductivity (EC), the UV absorbance and scanning electron microscope (SEM) data were used to study cellular morphology. The impact of ε-PL on cell metabolism was also investigated by different methods, such as enzyme activity (peroxidase [POD], catalase [CAT], succinodehydrogenase [SDH] and malic dehydrogenase [MDH]) and cell metabolic activity. The results showed that the MIC of ε-PL against *S. putrefaciens* was 1.0 mg/mL. When *S. putrefaciens* was treated with ε-PL, the growth of the bacteria was inhibited and the AKP content, electrical conductivity and UV absorbance were increased, which demonstrated that ε-PL could damage the cell structure. The enzyme activities of POD, CAT, SDH, and MDH in the bacterial solution with ε-PL were decreased compared to those in the ordinary bacterial solution. As the concentration of ε-PL was increased, the enzyme activity decreased further. The respiratory activity of *S. putrefaciens* was also inhibited by ε-PL. The results suggest that ε-PL acts on the cell membrane of *S. putrefaciens*, thereby increasing membrane permeability and inhibiting enzyme activity in relation to respiratory metabolism and cell metabolism. This leads to inhibition of cell growth, and eventually cell death.

## 1. Introduction

ε-Polylysine (ε-PL), a natural food preservative used commercially in Japan, has generally been recognized as safe [1]. Moreover, ε-PL has generally been recognized as a safe agent (GRAS No. 000135) by the Food and Drug Administration (FDA) and is commonly used in many foods, including soft drinks, cheese, egg-based dishes, salad dressings, fish, sauces and potato-based foods [1,2,3,4]. China has also approved it as a variety of food additive in April 2014. The main advantages of ε-PL is its water solubility, edibility and nontoxicity towards humans and the environment, as it can be degraded into lysine without any side effects for consumers [5]. Furthermore, ε-PL displays a broad spectrum of antimicrobial activity against most gram-negative and gram-positive bacteria, fungi and viruses [3]. Hence, ε-PL can be applied as a natural food additive in food products or as a disinfectant in dairy foods and their production equipment. A number of studies have demonstrated that ε-PL is effective in controlling foodborne bacteria, such as *Escherichia coli*, *Staphylococcus aureus* and *Bacillus subtilis* [6,7]. Although the antimicrobial activity of ε-PL and its application have been reported, its antimicrobial mechanism is vague and not thoroughly described.

*S. putrefaciens* has previously been shown to be one of the main spoilage organisms present in temperate fish species [8,9]. Nowadays, many studies have focused on antimicrobial techniques against *S. putrefaciens*. Schelegueda et al. [10] showed that the combined use of chitosan, nisin and sodium lactate inhibited the growth of *S. putrefaciens* by disrupting its outer membrane, affecting the ribosomes and DNA of the bacteria. Mu et al. [11] reported that 1% chitosan combined with 0.6% nisin had significant inhibitory effects on the growth and biofilm formation of *S. putrefaciens* and *Shewanella algae* in *Pseudosciaena crocea.* In addition, Fei et al. [12] showed that cinnamon oil combined with gamma radiation damaged the cell permeability and integrity of *S. putrefaciens*, while Zhang et al. [13] showed that *Ginkgo biloba* leaf extracts (GBLEs) damaged the morphology of *S. putrefaciens*.

Some researchers have shown that bacteriostatic agents could affect cell metabolism when they enter a bacterial cell after destroying its structure. Metabolic respiration produces cellular energy through the tricarboxylic acid cycle (TCA) and oxidative phosphorylation, which is used for cellular growth, multiplication, apoptosis and other cellular processes of living microorganisms [14]. The TCA cycle is not only the main method for gaining energy for the cell, but also the most common metabolic pathway for thorough oxidation of three main organic substances: sugar, fat, and protein. The respiratory system is one of the most common targets for antimicrobial agents. Additionally, defense enzymes play vital roles in host plant resistance against pathogen invasion, and reductions in disease severity in stressed plants have been attributed to changes in the activities of defense enzymes, such as peroxidase (POD) and catalase (CAT) [15].

Hence, the purpose of this research was to investigate the antibacterial mechanism of ε-PL against *S. putrefaciens* through (i) membrane leakage (alkaline phosphatase, electrical conductivity and ultraviolet absorption), (ii) mycelial morphology by scanning electron microscope (SEM) observation, (iii) enzyme activities (succinodehydrogenase (SDH), malic dehydrogenase (MDH), POD and CAT) in *S. Putrefaciens*, and (iv) cellular metabolism (viability) of the TCA pathway.

## 2. Results and Discussion

### 2.1. Minimal Inhibitory Concentration (MIC)

The MIC of ε-PL against *S. putrefaciens* was measured by conventional broth dilution assay, as depicted in Figure 1. OD_600 nm_ values decreased with an increase in ε-PL concentration, and the MIC of ε-PL against *S. putrefaciens* was found to be 1.0 mg/mL. Moreover, the inhibitory effect of ε-PL against *S. putrefaciens* was positively correlated with the concentration of ε-PL.

### 2.2. Growth Curve

The time courses of bacteria growth in the presence of MIC and 2 MIC ε-PL were plotted so as to further confirm the antibacterial activity of ε-PL against *S. putrefaciens*. As shown in Figure 2, bacterial growth in the control group (CK) followed an S-shaped growth curve, with *S. putrefaciens* reaching the logarithmic phase after 2 h and the stationary phase after 13 h. In addition, when ε-PL was added into the medium, the treated bacteria grew slower than the CK. Absorption values at 600 nm were lower than that of the control in almost each growth phase. These observed changes in the growth curves indicate antibacterial activity of ε-PL treatment, which suggests that ε-PL could delay the growth of *S. putrefaciens*.

### 2.3. Characterizations of the Cell Membrane

AKP is a kind of intracellular enzyme located between the cell wall and cell membrane; hence, its activity cannot be detected under normal conditions in the extracellular environment [16]. However, when the cell wall is damaged, AKP can leak out into the extracellular environment [17,18]. Therefore, the concentration of AKP in prepared cell suspensions may be used to reflect the integrity of bacterial cell walls.

As illustrated in Figure 3A, in the absence of ε-PL, AKP activity was maintained at 3.20 U/L. After treatment with MIC and 2 MIC ε-PL for 2 h, the AKP activity increased from 3.32 to 4.54 U/L and from 3.32 to 5.91 U/L, respectively. Obviously, the AKP activity from *S. putrefaciens* cells increased with an increase of ε-PL concentration (from MIC to 2 MIC). It has previously been reported that ε-PL induces cell wall damage, leading to AKP leakage from cells with the loss of cytoderm integrity [19]. Additionally, ε-PL can enter the inner membrane and lower the activity of AKP in *S. putrefaciens*, thus inhibiting bacterial growth. The decreased activity of AKP treated with ε-PL may prevent dephosphorylation progress, thereby inhibiting energy metabolism [15].

Electrical conductivity (EC) was examined to determine permeability changes in the cell membranes of the bacteria. As shown in Figure 3B, the EC values of the control were consistently lower than the ε-PL treated groups. With respect to the two ε-PL treated groups, the EC values of *S. putrefaciens* were larger for 2 MIC group. This increase in the EC of the treated bacterial suspension suggests the permeability of the cell membrane improved with ε-PL treatment, which may be caused by interactions between ε-PL and the cell membrane, thus leading to intracellular leakage of ions like Na^+^ and K^+^ [16].

The release of cell constituents corresponds to the integrity of the cell membrane. Small ions such as potassium and phosphate tend to flow out first, followed by macromolecular substances such as nucleotides [20]. Thus, these are good indicators for evaluating the integrity of the cell membrane [21]. The detection of absorbance at 260 nm could be used to estimate the amount of nucleotide leakage from the cytoplasm. The absorption of bacterial supernatant is shown in Figure 3C. Evidently, the absorbance at 260 nm increases immediately after 2 h of treatment by ε-PL, which indicates leakage of nucleotides from the *S. putrefaciens* cells into the extracellular environment. It may be speculated that damage to the outer membrane of *S. putrefaciens* is the main cause of release of intracellular components into the supernatant. These results indicate that ε-PL treatment results in damage to *S. putrefaciens* cells. The cell membrane is an important structural component of bacteria and a major site of action of various antibacterial agents [22]. When the cell membrane is damaged, the contents within the cell, such as some large molecules (nucleic acids) and ions (Na^+^, K^+^ and phosphate ions), will leak out. Accordingly, leakage of intracellular material is an important indicator of the integrity of the cell membrane.

### 2.4. Scanning Electron Microscope (SEM)

Morphology changes in the *S. putrefaciens* cells were evaluated by SEM analysis. Figure 4 shows SEM photomicrographs of *S. putrefaciens* cells treated with ε-PL.

As seen in Figure 4, the bacterial cells of *S. putrefaciens* without ε-PL had a regular, short-bar-shaped morphology, with a smooth and intact cell surface. Furthermore, the surface appeared full and glossy. In contrast to the control, *S. putrefaciens* cells treated with ε-PL (MIC and 2 MIC) were deformed, shriveled, pitted and swelled, with parts of the cell appearing broken and shrunk to a smaller size.

The results obtained by the SEM (Figure 5) testify to membrane damage and leakage of bacterial contents following treatment with ε-PL. Lin et al. [16] also reported that ε-PL could act on the membrane of *L. monocytogenes*, affecting the integrity of the membrane and inducing the loss of intracellular materials, enzymes and soluble proteins.

### 2.5. Effect of ε-PL on Respiratory Metabolism

#### 2.5.1. Measurement of ATPase Activities

ATP is produced in both the cell wall and cell membrane of bacteria via the glycolysis process. Bacterial membranes have multiple enzymes that can contribute to ATP generation and metabolism. ATPase is one of the most important enzymes for this mechanism [23]. ATPase catalyzes ATP to ADP, thus providing energy for cells. Since these enzymes provide cells with cofactors and energy, a decrease in their activity presumably impedes carbohydrate metabolism, which in turn retards cell growth or even leads to cell death [24]. This part of the study was performed to investigate the inhibition mechanism of ε-PL on the activity of ATPase. Figure 5 shows that the ATPase activity of the control sample was stable. On the contrary, the activity of ATPase exposed to ε-PL decreased remarkably. This decrement in the ATPase level of *S. putrefaciens* treated with ε-PL indicates that the ε-PL disrupted the cell membrane, inhibiting intracellular ATPase activity. This also led to the loss of the balance of external ATPase activity.

Lower ATPase activity may cause an outflow of ATP content from the bacteria, impeding its respiratory metabolism. Lin et al. [16] also found that ε-PL inhibited the respiration of *L. monocytogenes* by determining a change in its ATPase activity.

#### 2.5.2. Measurement of Peroxidase (POD) and Catalase (CAT) Activities

POD, a kind of oxidoreductase, and CAT, a kind of antioxidant, exist in all microorganisms. These enzymes correlate with microbial activity and respiration, maintaining the balance of active oxygen by converting H_2_O_2_ to H_2_O and O_2_^−^ [15]. They both exist widely in aerobic organisms and are an important part of the protective enzyme systems in a cell. POD and CAT can not only remove reactive oxygen radicals and prevent the destruction of a cell by physiochemical changes, they can also reduce the deposition of reactive oxygen in cells so as to ensure the integrity of the cell membrane and other structures. If their activity is inhibited, the operation of the whole enzyme system is greatly affected, and the protective function of the enzyme system is reduced or even lost. In the case of bacteria, a decrease of POD activity causes damage to the bacterial membrane system, which reduces the ability of the bacteria to resist external influences and ultimately affects the growth and metabolism of the microorganism.

As shown in Figure 6, the POD activity of *S. putrefaciens* treated with ε-PL decreased with an increase in processing time, while the normal solution remained relatively unchanged. Furthermore, the POD activity decreased obviously with the increase of ε-PL concentration. The tendencies of POD were similar to those of CAT. In summary, ε-PL inhibited CAT and POD activities of *S. putrefaciens*. Such results can be justified by the fact that POD and Superoxide Dismutase (SOD) activities may be generally considered as important in the resistance of *S. putrefaciens*.

POD and CAT are important protective enzymes in biological organisms and can scavenge oxidized free radicals in the body [25]. The lower the activity of POD and CAT, the weaker the cell’s ability to eliminate free radicals. These results show that ε-PL may prevent the growth of *S. putrefaciens* by hindering the transfer of information and energy, or by inhibiting the self-protection ability of the bacteria. Since these enzymes provide cells with cofactors and energy, decreases in their activities presumably impedes carbohydrate metabolism, which in turn retards cell growth or even leads to cell death.

#### 2.5.3. Measurement of Succinodehydrogenase (SDH) and Malicdehydrogenase (MDH) Activities

SDH is located in the inner mitochondrial membrane. This enzyme catalyzes succinic acid conversion to fumaric acid and simultaneously produces FADH_2_. SDH links oxidative phosphorylation to respiration metabolism, and thus is a representative enzyme in the mitochondria [24]. SDH activity in the treated cells decreased with increasing time, while it remained stable in the untreated cells (Figure 7). Compared with the control samples, the SDH activity of *S. putrefaciens* treated with ε-PL decreased. MDH is a representative enzyme in the tricarboxylic acid (TCA) cycle, which catalyzes L-malic acid to oxalacetic acid and simultaneously produces NADH [24]. Changes in MDH activity followed a similar pattern to that of SDH (Figure 7). After treatment with MIC and 2 MIC of ε-PL, the MDH activity of *S. putrefaciens* decreased. As expected, ε-PL treatment of *S. putrefaciens* resulted in significantly decreased activities of SDH and MDH in the bacteria, with the activity of the enzymes being inversely proportional to ε-PL concentration. These results were in agreement with Yang et al. [26], who observed lower SDH and MDH activity in *Penicillium italicum*.

All kinds of metabolism in living organisms are carried out under the catalysis of enzymes. The TCA cycle is not only the main way to gain energy for the cell, but it is also the common metabolic pathway for thorough oxidation of three main organic substances: sugar, lipids, and protein. Among all the enzymes in the TCA cycle, it seems that SDH links oxidative phosphorylation to respiration metabolism, and thus is a representative enzyme in the mitochondria. MDH widely exists in the mitochondria and bacterial cell membrane, and there are differences in its type in different organisms. MDH is closely related to the heredity, species category, growth and reproduction of cells or bacteria, and is an important enzyme in organisms [26]. In this study, ε-PL inhibited the activities of SDH and MDH, indicating retarded respiration, impeded cell viability, and even cell death. Studies have shown that in vivo Iodonitrotetrazolium chloride (INT) reduction may be used in marine bacteria to estimate respiration rate [27]. Through our research, the production rate of formazan was used to represent the metabolic activity of *S. putrefaciens* in the TCA cycle. With increasing concentration of ε-PL, the production rate of formazan gradually decreased, and the TCA of *S. putrefaciens* was blocked. In summary, these results suggest that ε-PL depressed the metabolic activities of *S. putrefaciens* by targeting key enzymes in the tricarboxylic acid (TCA) cycle.

#### 2.5.4. Effect of ε-PL on Cellular Metabolism (Viability)

The principle of determining the metabolic activity of cells by INT is that living cells can generate H^+^ via dehydrogenase activity during the TCA cycle. H^+^ can restore INT to a stable red formazan. The metabolic activity of bacteria is judged by the production rate of formazan. A reduction of OD_630 nm_ means a reduction in the production rate of formazan and thus weakened metabolism. In this work, with an increase in the concentration of ε-PL, the OD_630 nm_ was found to decrease significantly, with its inhibitory effect becoming stronger (Figure 8). Thus, the cellular metabolism (viability) of *S. putrefaciens* was inhibited by ε-PL.

## 3. Materials and Methods

### 3.1. Bacterial Culture

*S. putrefaciens* were separated and purified from a large yellow croaker (*Pseudosciaena crocea*), identified and preserved by Shanghai Engineering Research Center of Aquatic Product Processing and Preservation.

### 3.2. Determination of Minimal Inhibitory Concentrations (MICs)

MIC was measured by referring to the conventional broth dilution method of Li et al. [6] with some modifications. *S. putrefaciens* were incubated at 37 °C for 8–10 h to approximately 10^6^–10^7^ CFU/mL in Tryptic Soy Broth (TSB) medium. ε-PLs of serial dilutions, previously dissolved in sterile distilled water, were prepared to final concentrations of 0, 0.25, 0.5, 1, 2, 4, 8, 16 and 32 mg/mL of TSB medium. After 18–24 h shake incubation (150 rpm at 37 °C), MIC was determined as the lowest concentration of ε-PL which visibly inhibited microorganism growth, which was confirmed by measuring the OD_600 nm_ of all treatments with a spectrophotometer (V-1100, Meipuda Instrument Co. Ltd., Shanghai, China). The sample without ε-PL was set as the control.

### 3.3. Bacterial Growth Curve

The inhibition effect of ε-PL on the growth of *S. putrefaciens* was evaluated. In brief, the logarithmic phase-tested bacteria were harvested and adjusted to 1.0 × 10^7^ CFU/mL with TSB broth. One mL of the bacterial suspension was inoculated into the flasks containing 100 mL fresh sterile TSB broth. ε-PL was added to the culture to keep a final concentration of MIC and 2 MIC. The tested bacteria culture without ε-PL was used as the control. All cultures were incubated on an orbital shaker (150 rpm at 37 °C). The growth rates and bacterial concentrations were monitored every hour by measuring the OD_600 nm_ values using a Microplate Reader (Bioscreen Co., Helsinki, Finland).

### 3.4. Effects of ε-PL on the Cell Wall and Membrane Permeability of S. putrefaciens

*S. putrefaciens* were cultured to the logarithmic phase before treatment with MIC and 2 MIC ε-PL. Samples without ε-PL were set as the control samples. All cultures were incubated on an orbital shaker (150 rpm at 37 °C). Aliquots of the samples were drawn at regular intervals (2 h) and centrifuged (3500 r/min, 10 min).

#### 3.4.1. Alkaline Phosphate (AKP) Activity

The effects of ε-PL on AKP were studied using the AKP assay kit, purchased from the Nanjing Jiancheng Bioengineering Institute (Jiangsu, China). The AKP levels were measured using the kits in line with the manufacturer’s instructions. Changes in absorbance at 520 nm were measured to represent the activity of AKP.

#### 3.4.2. Electrical Conductivity (EC) Measurement

Changes in the EC of the surface of the cell membranes were recorded by a portable multi-parameter analyzer (DZS-718, Shanghai Precision Science Instrument Co., Ltd., Shanghai, China).

#### 3.4.3. Determination of Nucleic Acid Leakage (OD_260nm_)

The measurement of the leakage of nucleic acid was carried out using the method reported by Wang et al. [28] with some modifications. Firstly, the bacterial suspension was washed three times with distilled water by centrifugation (6000 rpm for 10 min at 4 °C). The suspensions were then prepared to contain 10^7^ CFU/mL in 0.9% sterile NaCl. Aliquots of the samples were drawn at regular intervals (1 h) and centrifuged. The absorbance of 260 nm UV-absorbing materials in the suspension was then measured by an ultraviolet–visible light detector (Unico Instrument Co. LTD, Shanghai, China).

### 3.5. Scanning Electron Microscope (SEM) Observations

The prepared 1% bacterial solution of *S. putrefaciens* was inoculated into the ε-PL solution. The inoculations were incubated at 37 °C with shaking (150 rpm) for 12 h. The bacterial cells were collected by centrifugation at 8000 r/min for 5 min at 4 °C. The bacteria precipitations were fixed in 2.5% glutaraldehyde solution at 4 °C for 10 h. The fixed bacteria cells were washed three times with phosphate buffer solution (PBS), followed by serial dehydration with 30%, 50%, 70%, 90% and 100% ethanol solutions at 1 min intervals. After 12 h of freeze-drying, the cells were covered through cathodic spraying, after which SEM observations were performed.

### 3.6. Effect of ε-PL on Respiratory Metabolism

#### 3.6.1. Measurement of Adenosine Triphosphatase (ATPase)

The effects of ε-PL on ATPase were studied using an ATPase assay kit, purchased from the Nanjing Jiancheng Bioengineering Institute (Jiangsu, China). The ATPase levels were measured using the kits in line with the manufacturer’s instructions. Changes in absorbance at 636 nm were taken to function as the ATPase level.

#### 3.6.2. Measurement of Peroxidase (POD) and Catalase (CAT) Activities

The POD and CAT activities of *S. putrefaciens* cells with ε-PL at various concentrations (0, MIC and 2 MIC) were determined using commercially available spectrophotometry kits (Nanjing Jiancheng Bioengineering Institute, Jiangsu, China) following the manufacturer’s instructions. POD and SDH activities were determined at 420 and 405 nm, respectively.

#### 3.6.3. Measurement of Succinodehydrogenase (SDH) and Malicdehydrogenase (MDH) Activities

The SDH and MDH activities of *S. putrefaciens* with ε-PL at various concentrations (0, MIC and 2 MIC) were determined using commercially available spectrophotometry kits (Jiancheng Bioengineering Institute, Jiangsu, China) following manufacturer’s instructions. MDH activities were estimated at 340 nm in terms of the redox reaction and SDH activities were determined at 600 nm.

### 3.7. Effect of ε-PL on Cellular Metabolism (Viability)

The inhibition effect of ε-PL on cellular metabolism (viability) of *S. putrefaciens* was evaluated according to [29] with slight modifications. Bacteria in the logarithmic phase were harvested by centrifugation at 5000 r/min for 10 min. The concentration of *S. putrefaciens* was adjusted to 10^6^ CFU/mL by washing and resuspending with saline. In order to achieve the final concentrations of 0, 1 and 2 MIC, different volumes of ε-PL were added to a bacterial suspension. After incubation at 37 °C for 1 h, the mixture was centrifuged at 10,000 r/min for 10 min and the thallus was resuspended with 0.9% saline. Iodonitrotetrazolium chloride (INT, 1 mmol/L final concentration) was added to the solution, followed by incubation conducted at 37 °C for 30 min. The maximum absorbance of formazan was measured at 630 nm as an estimate of the cellular metabolism (viability).

### 3.8. Statistical Analysis 

The experiment followed a completely randomized design (*n* = 3). Analyses were performed by SPSS 16.0 software package and curves were drawn by Origin 8.6. One-way analysis of variance (ANOVA) and Student’s t-tests were used to elucidate significant differences at a significance level of *p* < 0.05.

## 4. Conclusions

From our findings, we suggest ε-PL may inhibit *S. putrefaciens* through the following aspects. The Schematic representation of the possible inhibitory mechanisms of ε-PL against *S. putrefaciens* was as shown in Figure 9. Firstly, ε-PL may affect the cell structure of *S. putrefaciens* and cause irreversible damage to the cell membrane. Evidence of leakage of some large molecules (nucleic acids), ions (Na^+^ and K^+^) and intracellular enzymes (AKP and ATPase) supports this suggestion. Besides, ε-PL appears to be capable of damaging bacterial morphology. Secondly, our results indicate that ε-PL inhibits the physiological activity of *S. putrefaciens* via its effect on defense enzymes (POD and CAT) of the respiratory system and key enzymes (SDH and MDH) of the TCA cycle. ε-PL appears to decrease the TCA cycle and biosynthesis of *S. putrefaciens*, which can lead to bacterial decomposition and eventual death. Further work is also needed to test ε-PL against *S. putrefaciens* by Embden-Meyerhof-Parnas (EMP) and Hexose Monophosphate Pathway (HMP).

## Figures and Tables

**Figure 1 molecules-24-03727-f001:**
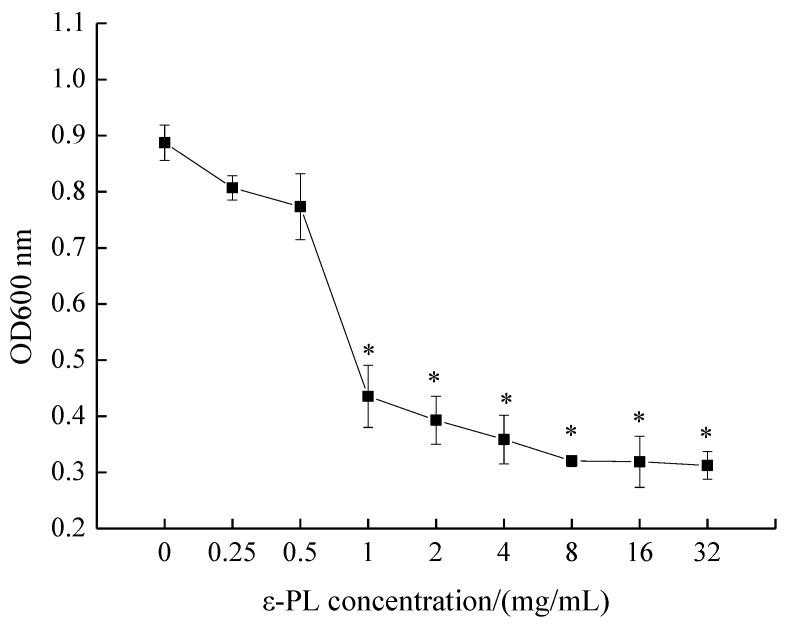
Minimal inhibitory concentration (MIC) of ε-polylysine (ε-PL) against *S. putrefaciens.* Bars represent the standard error of the means. The superscript (*) indicates a result that is significantly different to the equivalent control point (*t* test, *p* < 0.05).

**Figure 2 molecules-24-03727-f002:**
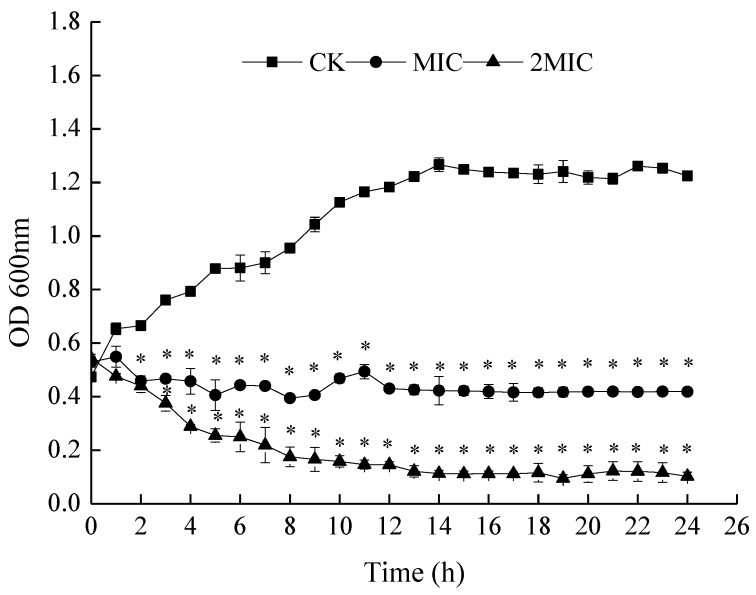
Effect of ε-PL on the growth curve of *S. putrefaciens.* Bars represent the standard error of the means. The superscript (*) indicates a result that is significantly different to the equivalent control point (*t* test, *p* < 0.05).

**Figure 3 molecules-24-03727-f003:**
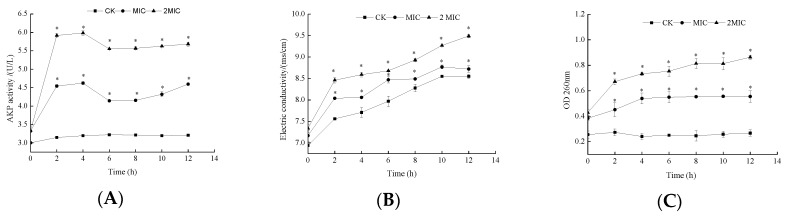
Effects of ε-PL on AKP (**A**), Electrical conductivity (**B**) and OD 260 nm (**C**) of *S. putrefaciens.* Bars represent the standard error of the means. The superscript (*) indicates a result that is significantly different to the equivalent control point (*t* test, *p* < 0.05).

**Figure 4 molecules-24-03727-f004:**
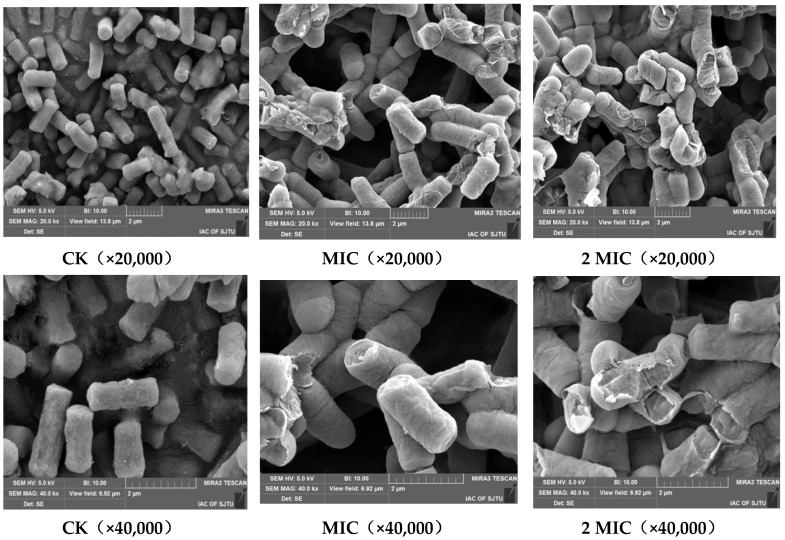
Scanning electron microscope (SEM) images of *S. putrefaciens* treated with sterile water (CK), 1 mg/mL ε-PL (1 MIC) and 2 mg/mL ε-PL (2 MIC).

**Figure 5 molecules-24-03727-f005:**
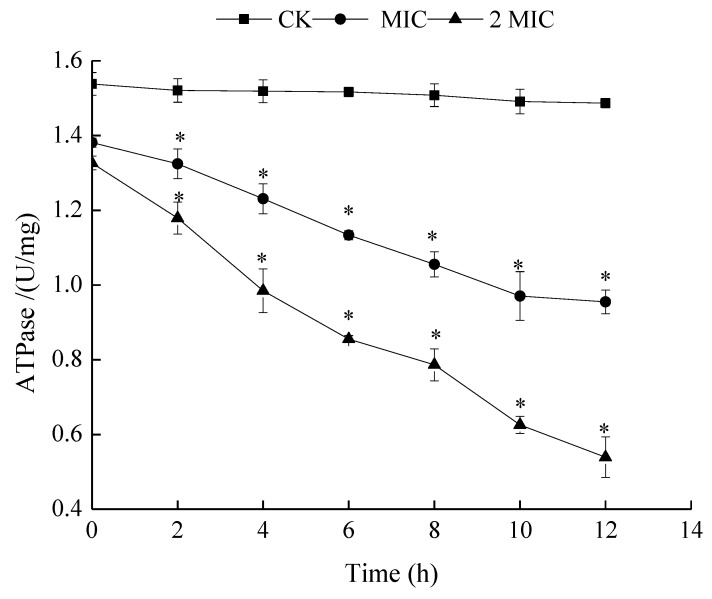
Effects of ε-PL against on ATPase activity of *S. putrefaciens*. Bars represent the standard error of the means. The superscript (*) indicates a result that is significantly different to the equivalent control point (*t* test, *p* < 0.05).

**Figure 6 molecules-24-03727-f006:**
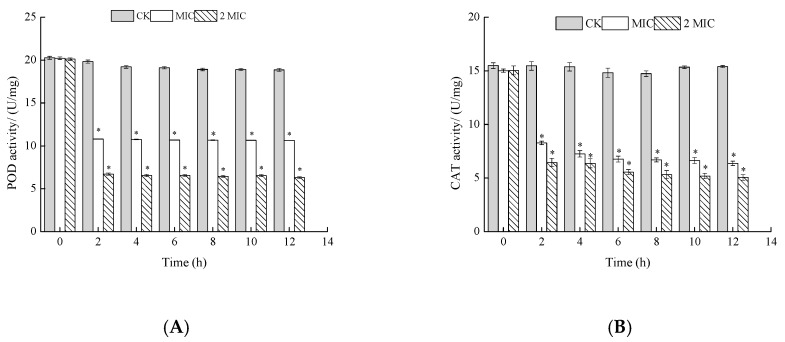
Effects of ε-PL on peroxidase (POD) (**A**) and catalase (CAT) (**B**) activity of *S. putrefaciens.* Bars represent the standard error of the means. The superscript (*) indicates a result that is significantly different to the equivalent control point (*t* test, *p* < 0.05).

**Figure 7 molecules-24-03727-f007:**
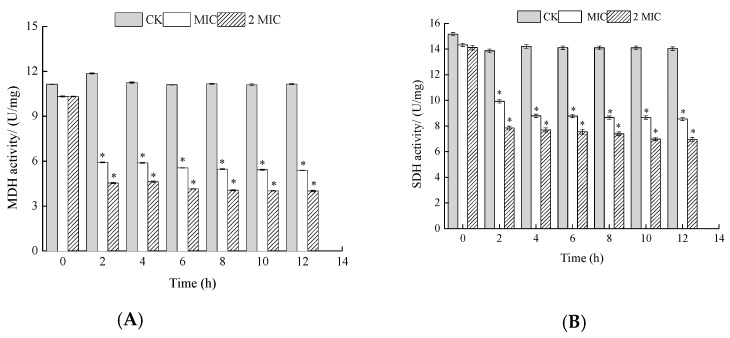
Effects of ε-PL on malicdehydrogenase (MDH) (**A**) and succinodehydrogenase (SDH) (**B**) activity of *S. putrefaciens.* Bars represent the standard error of the means. The superscript (*) indicates a result that is significantly different to the equivalent control point (*t* test, *p* < 0.05).

**Figure 8 molecules-24-03727-f008:**
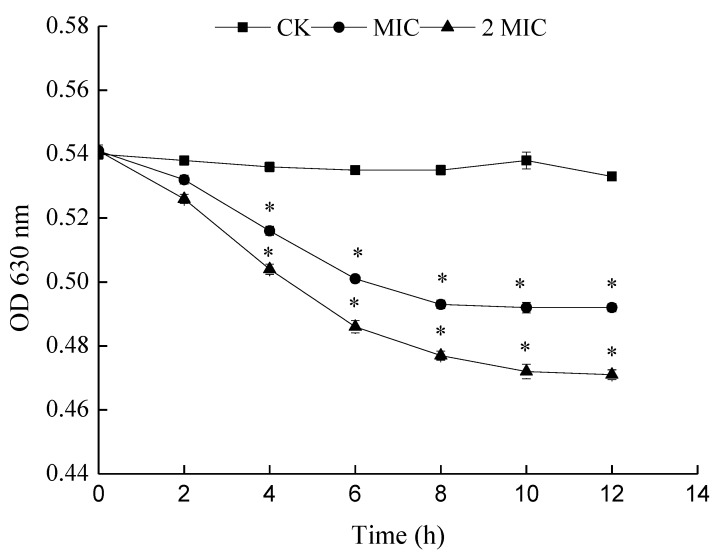
Effects of ε-PL on cell metabolism of *S. putrefaciens*. Bars represent the standard error of the means. The superscript (*) indicates a result that is significantly different to the equivalent control point (*t* test, *p* < 0.05).

**Figure 9 molecules-24-03727-f009:**
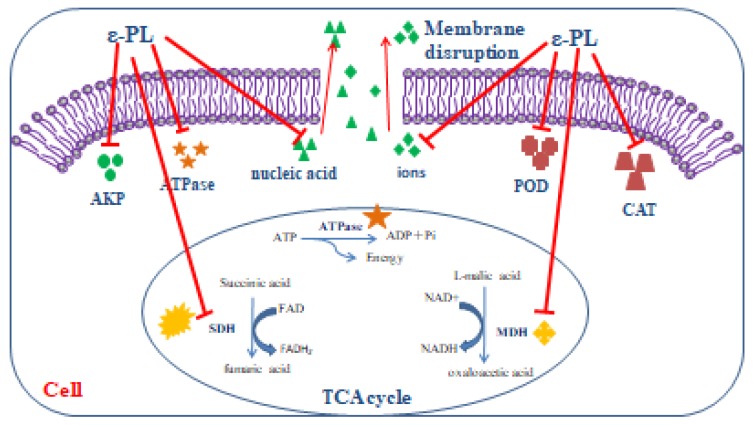
Schematic representation of the possible inhibitory mechanisms of ε-PL against *S. putrefaciens*.

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
