# Peer review of "ε-Polylysine Inhibits Shewanella putrefaciens with Membrane Disruption and Cell Damage"

_molecules, 2019, doi:10.3390/molecules24203727_

Round 1
Reviewer 1 Report
The aims of this study was intended to investigate the antibacterial mechanism of ε-PL against Shewanella Putrefaciens through (i) membrane leakage (alkaline phosphatase, electrical conductivity and ultraviolet absorption), (ii) the mycelial morphology by scanning electron microscopy observation (SEM), (iii) enzyme activities (SDH, MDH, POD and CAT) in S. Putrefaciens, and (iv) cellular metabolism viability of TCA pathway. This study is ver interesting and well design. It could be recommended for publication on Molecules after major revision. Major points are addressed as follows,
How did authors calculate the MIC? Which statistical method was used in this study? All figures should be marked star if the statistical analysis was significance.Author Response
How did authors calculate the MIC? Which statistical method was used in this study? All figures should be marked star if the statistical analysis was significance.
Response:
1. How did authors calculate the MIC?
As described in this article (2.2), MIC was determined as the lowest concentration of ε-PL which visibly inhibited the growth of microorganism.
2. Which statistical method was used in this study?
The method was used in this study is according to the result of OD600nm from each group.
3. All figures should be marked star if the statistical analysis was significance.
We have corrected the figures according to the Reviewer’s suggestion. Special thanks to you for your good comments.
Reviewer 2 Report
Two general comments:
I - Latin name of Shewanella putrefaciens should be written as S. putrefacines in abbreviated form after the first mention in text. Correct this through a whole Manuscript.
II - You did not follow Instructions for authors for this Journal regarding to Reference citing in text and References list at the end of Manuscript. Check all and correct.
Specific comments:
All other comments are given with an appropriate Line number(s) from text in order to keep it easier to follow:
Lines 5 -10: Why brackets in authors affiliations? Explain or correct.
Lines 15-16: Suggest to split in two sentences starting with :"The membrane permeability..." in Line 15. It will be easier to read and understand.
Line 18: Add "different" after "by".
Lines 21-22: Suggest to reorder words as follow: "When S. putrefacines was treated with ε-PL...."
Line 22: Delete , after "values" and replace with "and". Also, word Absorbance (Line 23) should not be written with capital letter B.
Line 25: Finish the sentence after:"bacterial solution.". Start the new one as:" The higher..."
Line 29: delete "metabolism" after "respiratory". It is surplus.
Line 30: Delete "killed" and add "death" after "cell". It is more appropriate here.
Lines 34-35: Reorganize sentence as follow:"ε-poly-lysine (ε-PL), a natural food preservative, commercially used in Japan (1) has been generally recognized as a safe (2)."
Line 42: Add "of" after "spectrum".
Line 54: "nisin" should be written with a small capital letter.
Line 56: The same as previous for "chitosan" and "nisin".
Line 57: give the full name for P. crocea since this is the first time that it is mentioned.
Line 88: Add "determined as" after "MIC was".
Line 89: Add "nm" in subscript of "OD600" i.e. "OD600nm" as you done in Line 99.
Lines 90-91: Replace "of no adding" with "without". It is much more appropriate here.
Line 117: There is no need for , in "Wang, et al.,". It should be written as "Wang et al." Correct the same errors in Lines 234, 256 and 298.
Line 120: "the" not "The".
Line 121: "was" instead of "were".
Line 124: Suggest to re-arrange as:"The prepared 1% bacterial solution..."
Line 125: Always split units (except in case of &) with "one space" from numerical value. There are several errors through a whole Manuscript. So, please check all and correct.
Line 169: Suggest to replace "positive correlation" with "positively correlated". It seems more suitable here.
Line 175: Add "by" after "followed" and "of" in front of "S-shaped".
Line 178: Move word "almost" after "in".
Lines 178-180: Can you , please, rephrase this sentence. It is unclear what did you mean by this "could be shown"?
Line 184: Delete "and". It is surplus.
Line 187: "is damaged" not "was damaged". Also, delete "and".
Lines 191-193: Values given in text are not in agreement with values on Figure 3. For instance, the lowest value visible on Figure 3-a is about 3 and in text you are talking about value lower than 2. Please, check and correct what is necessary. In addition, at Figure 3 you should add marks a), b) and c) since you mentioned them in text.
Lines 197-200: This part of text is not in agreement with your previous results and explanation.Please, check and correct/improve what is necessary.
Line 202: Conductivity is not given in Figure 3-b but 3-c. Also, there is no rapid increase of EC value just increase. So, delete word "rapidly" in Line 203.
Line 206: Add "to" after "leading".
Line 213: Absorption is not given in Figure 3-c but 3-b.
Line 215: Add "cells" after "S. putrefacines".
Line 221: Since you are previously define phosphate ion as the small one I suggest to add it here along with Na+ and and K+ ions.
Line 231: In description of Figure 4 I suggest to add "swelled" since it is how I will describe cells. Especially in case of MIC.
Line 242: Something is missing here when you describe ATPase action.Something like "ATP conversion in ADP" or "dephosphorylation of ATP" or something similar.
Line 247-248: You determine ATPase activity not content. Please correct.
Line 250: Add "of" after "lose".
Lines 254-255: Delete this sentence. It is surplus since you already said that (Lines 242-243).
Line 262: "exist" not "existed" as well as "are" not "were". It is something that is permanent.
Lines 273-274: Please, rephrase sentence "POD activity...". It is unclear.
Line 276: Add "as" after "considered".
Line 288: Add "conversion" after "acid". Replace "of" with "in".
Line 293: Add some description for this process. Like "transformation" or something like that.
Lines 302-303: Here you are talking about classes (plural) of organic molecules so it must be in plural: sugars, lipids (instead of fat) and proteins.
Lines 304-305: Delete sentence "SDH catalyzes..." It is already given in Lines 287-288.
Line 312: Abbreviation INT was not define previously in text. So, please, define here.
Lines 315-318: Here you are talking about formazan measurement. But you did not give any results for that on some Figures or Tables? Can you, please, check/correct/improve this.
Line 321: Please delete "a kind of". There is only one H+ ion type.
Line 337-338: Further work is not "possible" but "needed". Correct this.
Author Response
I - Latin name of Shewanella putrefaciensshould be written as putrefacines in abbreviated form after the first mention in text. Correct this through a whole Manuscript.Response: We have corrected this part according to the Reviewer’s suggestion. Special thanks to you for your good comments.
II - You did not follow Instructions for authors for this Journal regarding to Reference citing in text and References list at the end of Manuscript. Check all and correct.Response: We have corrected this part according to the Reviewer’s suggestion. Special thanks to you for your good comments.
Specific comments:
Lines 5 -10: Why brackets in authors affiliations? Explain or correct.Response: The affiliations of authors were correct, because the affiliations were one of the parts in Shanghai Ocean University.
Lines 191-193: Values given in text are not in agreement with values on Figure 3. For instance, the lowest value visible on Figure 3-a is about 3 and in text you are talking about value lower than Please, check and correct what is necessary. In addition, at Figure 3 you should add marks a), b) and c) since you mentioned them in text.Response: Considering the reviewer’s suggestion, I have checked the sequence of your figure and its correspondence in the context and corrected mistakes.
Lines 197-200: This part of text is not in agreement with your previous results and explanation. Please, check and correct/improve what is necessary.Response: Considering the reviewer’s suggestion, I have checked the sequence of your figure and its correspondence in the context and corrected mistakes.
Line 213: Absorption is not given in Figure 3-c but 3-b.Response: Considering the reviewer’s suggestion, I have rearranged the order of the figures.
Line 242: Something is missing here when you describe ATPase action. Something like "ATP conversion in ADP" or "dephosphorylation of ATP" or something similar.Response: Considering the reviewer’s suggestion, we have added the role of ATPase in the cell.
Line 293: Add some description for this process. Like "transformation" or something like that.Response: Considering the reviewer’s suggestion, we have added the role of SDH in the cell.
Line 312: Abbreviation INT was not defined previously in text. So, please, define here.Response: The INT was already defined at the method in 2.7, that is “The iodonitrotetrazolium chloride (INT, 1mmol/L final concentration) was added to the solution”.
Lines 315-318: Here you are talking about formazan measurement. But you did not give any results for that on some Figures or Tables? Can you, please, check/correct/improve this. Response: The reduction of OD630nm means the production rate of formazan and the metabolism was weak at a higher ε-PL concentration.
Response:
All the results were discussed and some of them were compared with other works. We acknowledge the reviewer’s comments and suggestions very much, which are valuable for improving the quality of our manuscript. We sincerely hope this manuscript will be acceptable to be published on “molecules” finally and look forward to hearing from you soon.
Thank you very much!
Best regards,
Yours sincerely,
First Author:
Wei-qing LAN
wqlan@shou.edu.cn
Corresponding Author:
Jing XIE
jxie@shou.edu.cn

Round 2
Reviewer 1 Report
Which statistical method was used? What criteria was set when statistic was significance?
Some Figures also could not mark symbol of statistical significance.
Author Response
1. Which statistical method was used? What criteria was set when statistic was significance?
Response:
SPSS 16.0 was applied to analyze one-way analysis of variance of data. Student's t-test were used to determine the significant differences at a significance level of P < 0.05.
2. Some Figures also could not mark symbol of statistical significance.
Response:
We have corrected this part according to the Reviewer’s suggestion.
3. The instructions References should be listed with numbers in text instead of author's name?
Response:
We have corrected this part according to the Reviewer’s suggestion. Special thanks to you for your good comments.
4. Due to the limitation of QQ email addresses (1253214783@qq.com), please provide us some alternative ones.
Response:
The alternative email addresses was 15836066261@163.com;
5. Other modifications.
Response:
Because professor Shu-cheng LIU (E-mail: Lsc771017@163.com ) and Graduate student named as Meng-ling CHEN (E-mail: chenml5211314@126.com) have contributed to this research, the name of other two authors have been added in this paper. The author's affiliations name as "College of Food Science & Technology, Guangdong Provincial Key Laboratory of Aquatic Product Processing and Safety, Guangdong Ocean University, Zhanjiang 524088" was added in this paper. The information of corresponding author, "Shu-cheng Liu. Tel.: 86-759-2396029; fax: 86-759-2396029; mail address: Lsc771017@163.com", was added in this paper. The information of acknowledgements was modified, that is" National Natural Science Foundation of China (grant number: 31972142), China Agriculture Research System (CARS-47-G26), Guangdong Provincial Key Laboratory of Aquatic Product Processing and Safety (GDPKLAPPS1802), Ability promotion project of Shanghai Municipal Science and Technology Commission Engineering Center (19DZ2284000)"All the results were discussed and some of them were compared with other works. We acknowledge the reviewer’s comments and suggestions very much, which are valuable for improving the quality of our manuscript. We sincerely hope this manuscript will be acceptable to be published on “molecules” finally and look forward to hearing from you soon.
Thank you very much!
Best regards,
Yours sincerely,
First Author: Wei-qing Lan (wqlan@shou.edu.cn)
Corresponding Author: Jing XIE, jxie@shou.edu.cn; Shu-cheng LIU, Lsc771017@163.com

Reviewer 2 Report
No further comments.
Author Response
Dear editors and reviewers:
Thank you for your reply and for the reviewers’ comments concerning our manuscript entitled “ε-polylysine inhibits Shewanella putrefacines with membrane disruption and cell damage”. Those comments are all valuable and very helpful for revising and improving our paper, as well as the important guiding significance to our research. We have discussed with these comments carefully and made correction which we hope meet with approval. We highlighted the revised portion in the paper. The main corrections in the paper and the responds to the reviewer’s comments are as follows:
Responds to the reviewer’s comments:
The instructions References should be listed with numbers in text instead of author's name?
Response:
We have corrected this part according to the Reviewer’s suggestion. Special thanks to you for your good comments.
Thank you very much!
Best regards,
Yours sincerely,
First Author: Wei-qing Lan (wqlan@shou.edu.cn)
Corresponding Author: Jing XIE, jxie@shou.edu.cn; Shu-cheng LIU, Lsc771017@163.com
